# Effect of Epidermal Growth Factor in Human Milk and Maternal Diet on Late-Onset Breast Milk Jaundice: A Case-Control Study in Beijing

**DOI:** 10.3390/nu14214587

**Published:** 2022-11-01

**Authors:** Qianying Guo, Mingxuan Cui, Xinran Liu, Shilong Zhao, Peng Liu, Linlin Wang

**Affiliations:** 1Department of Clinical Nutrition, Peking University People’s Hospital, Beijing 100044, China; 2Institute of Reproductive and Child Health/National Health Commission Key Laboratory of Reproductive Health, School of Public Health, Peking University, Beijing 100191, China

**Keywords:** late-onset breast milk jaundice, breast milk, postpartum diet, epidermal growth factor

## Abstract

Breast milk is crucial in the development of late-onset breast milk jaundice (BMJ), possibly due to the composition of breast milk and the lactating mother’s diet. To explore the possible nutritional pathogenesis of late-onset BMJ, we investigated the lactation diet and collected breast milk by following the 42-day postpartum mother–infants pairs in Beijing and a total of 94 pairs were enrolled. The macronutrient content of breast milk was measured, and the epidermal growth factor (EGF) content in breast milk was determined by ELISA. Data on in-hospital and out-of-hospital breastfeeding, infant growth, jaundice-related vaccination, and puerperium diet were collected. The BMJ group received the second dose of hepatitis B vaccine later than the control group, and the difference was statistically significant (*p* < 0.001). The EGF concentration in breast milk was lower in the BMJ group than in the control group (*p* = 0.03). When EGF increased by 1 ng/mL, the transcutaneous bilirubin (TcB) value decreased by 0.33 ng/mL and 0.27 ng/mL before and after the adjustment, respectively. A 1 g increase in oil intake led to a 0.38 ng/mL increase in EGF concentration before the adjustment. With a 1 g increase in oil intake, the TcB value decreased by 0.27 ng/mL before the adjustment, and with a 1 g increase in soybean and soybean product intake, the TcB value decreased by 0.34 ng/mL after the adjustment. Collectively, EGF in breast milk may inhibit the occurrence of late-onset BMJ, and the dietary intake of oil in lactating mothers may affect the level of EGF in breast milk, thus affecting the occurrence of late-onset BMJ. Finally, dietary oil intake may be a protective factor for the occurrence of late-onset BMJ by increasing EGF levels in breast milk.

## 1. Introduction

The World Health Organization (WHO) recommends exclusive breastfeeding for the first 6 months of life [1], but breastfeeding rates are suboptimal. Statistics in 2016 showed that breastfeeding rates at 6, 12, and 24 weeks postpartum in Beijing were 75.6%, 68.9%, and 53.2%, respectively [2]. Among the factors causing interruption or early termination of breastfeeding, breast milk jaundice (BMJ) is more common [3], especially late-onset BMJ, which occurs 7–10 days after birth and lasts for 6–12 weeks [4]. BMJ is a type of jaundice that occurs in newborns due to breastfeeding [3]. Owing to the abnormal accumulation of bilirubin, it causes the newborn’s skin to turn yellow, manifesting as jaundice. Most studies have shown that exclusive breastfeeding is associated with improved brain development and cognitive performance [5,6], which can provide protection for the immature brain against the effects of jaundice [7]. Although clinicians generally advise continuing exclusive breastfeeding during BMJ, the impact of neonatal BMJ on reduced breastfeeding and vaccination rates has been widely observed in the community, which in turn may have adverse consequences for infant growth and disease prevention [8,9]. Therefore, clarifying the pathogenesis of late-onset BMJ has important public health significance for preventing the occurrence of the above situations.

It has been suggested that bioactive factors in breast milk, such as epidermal growth factor (EGF) [10], may be associated with the development of late-onset BMJ. First discovered in rodents, EGF is found in a wide range of human tissues and body fluids, including blood, milk, gastric juice, and amniotic fluid. EGF has a variety of biological effects, including promoting intracellular DNA, RNA, and protein synthesis; stimulating cell proliferation and differentiation; inhibiting gastric acid secretion; promoting the synthesis of fruit acids and prostaglandins; and regulating the role of sperm and ovarian development and reproductive function [11]. Previous studies [12] indicated that EGF in breast milk can promote the development of the neonatal gastrointestinal tract. A study by Kumral et al. [10] found that the severity of BMJ was associated with increased EGF in breast milk.

The diet of lactating mothers has a greater influence on the composition of breast milk, and the intake of total dietary energy, protein, carbohydrate, fat, vitamin, and mineral all have an impact on the macronutrient content of breast milk [13]. Few studies have reported a relationship between breast milk EGF content and dietary intake. Lu M et al. [14] found that in addition to grains and meat, other dietary components, including fruits, vegetables, dairy products and soy products, were negatively correlated with EGF content in breast milk.

The aim of this nested case-control study was to promote breastfeeding and elucidate the pathogenesis of late-onset BMJ. We hypothesized that EGF in breast milk has an effect on late-onset BMJ and that dietary intake can affect the concentration of EGF, leading to alterations in transcutaneous bilirubin (TcB). To test this hypothesis, we conducted a nested case-control design to investigate whether mothers’ diet and EGF in breast milk affect the development of late-onset BMJ and to explore the possible nutritional strategies that improve EGF levels in breast milk.

## 2. Materials and Methods

### 2.1. Study Design and Study Subjects

Healthy, full-term, exclusively breastfed or predominantly breastfed (with more than 70% breastfed) newborns born at Peking University People’s Hospital between October 2020 and July 2021 were eligible for enrollment. Neonates with gestational age less than 37 weeks or risk factors such as blood group incompatibility, positive Coombs’ test, glucose-6-phosphate dehydrogenase deficiency, hemolytic disease, reticulocytosis, abnormal blood smear, erythrocytosis, cephalohematoma, history of asphyxia, hypothermia, intracranial hemorrhage, and cholestasis were excluded [15]. Newborns were also excluded from the study if their mothers had severe liver or kidney disease, psychological disorders, AIDS, hepatitis B, or other infectious diseases. Neonates who did not meet the above exclusion criteria and did not meet the following BMJ diagnostic criteria were included in the control group. Those who met the diagnostic criteria for late-onset BMJ were included in the BMJ group [16]: (i) Full-term infants, exclusively breastfed or mainly breastfed. (ii) Jaundice occurs 1 week after birth, and the serum total bilirubin value exceeds the physiological range. The degree of jaundice is mainly mild to moderate, with a peak at 2~3 weeks. (iii) Serum bilirubin decreased by about 30% after 1~3 days of breastfeeding. (iv) Detailed medical history, physical examination, and necessary examinations were taken and other possible pathological jaundice was excluded. A total of 29 jaundiced pairs (TcB value ≥ 7.87 mg/dL at 42 days after delivery) and 65 controlled pairs (TcB value < 7.87 mg/dL at 42 days after delivery) [16,17] were eventually enrolled (Figure 1). The study was approved by the Ethics Committee of the Peking University People’s Hospital, and parents were informed and agreed upon before enrollment.

### 2.2. Data Collection

Demographic data, including maternal age, height, weight, BMI, gestational age, date of delivery, mode of delivery, parity, gender, and weight of newborn, were collected.

The data of lactation in hospital and growth and feeding of 42-day infants were collected by questionnaire survey, including initial time of lactation, time to start breastfeeding, weight loss, and hospitalization time of puerperae, as well as body length, weight, breastfeeding frequency, bowel frequency, and the second dose of hepatitis B vaccine postponement time of 42-day infants.

A food frequency questionnaire (FFQ) containing food items such as milk, cereal, tubers, soybean and soybean products, vegetable, fruit, meat, egg and oil was used to obtain information on nutritional behavior over a 42-day period, and the amount of each food consumed by the lactating mothers was calculated according to the China Food Composition Tables: Standard Edition [18].

### 2.3. Milk Specimen Collection, Processing and Milk EGF Assays

Between 9 a.m. and 12 a.m. at 42 days postpartum, 5 mL of milk from unilateral midbreastfeeding (5–7 min after breastfeeding) was collected by hand milking with a sterile tube or manual breast pump. Samples were frozen to −80 °C for testing. A breast milk composition analyzer (HKANGYU KY-9003) was used for the detection of fat, protein, lactose, water, energy, and density in breast milk.

The EGF concentration in breast milk was determined using an enzyme-linked immunosorbent assay (ELISA) kit (Bioss BSK11025). The breast milk specimens were removed from the refrigerator at −80 °C, thawed at room temperature, dispensed into sterile tubes, and centrifuged for 20 min (3000 rpm) to collect the supernatant. A monoclonal antibody specific for EGF was precoated on the microplate. The standards and samples are pipetted into the wells, and any EGF present is bound to the immobilized antibody. After incubation, the unbound sample was removed in the washing step, and the EGF specific detection antibody was added to the well and combined with the combination of capture antibody-EGF in the sample. Unbound conjugates were removed after cleaning, and enzyme conjugates were added to the wells. The substrate is added after the incubation and washing steps. A colored product is formed in proportion to the EGF content in the sample. The reaction was terminated by the addition of acid, and the absorbance was measured at 450 nm. By measuring the concentration of EGF samples, standard curves of 7 standard dilutions were obtained.

### 2.4. Statistical Analysis

The data were statistically analyzed using SPSS version 24.0 (SPSS, Chicago, IL, USA). Scientific graphs were drawn using GraphPad Prism software version 8.0. Continuous variables with normal distributions were described by the mean ± standard deviation, and a *t*-test was used for comparisons between groups. Categorical data are expressed as percentages, and comparisons between groups were performed using a χ^2^ test. The relationship between dietary type, EGF content, and late-onset BMJ was analyzed by multiple linear regression (stepwise) before (Model 1) and after (Model 2) adjusting for confounding factors, including maternal age, BMI, gestational age, bowel frequency, breastfeeding frequency, pregnancy complications, newborn birth weight, newborn gender, 1-min Apgar score, 5-min Apgar score, initial time of lactation, time to start breastfeeding and hospitalization time. The results were evaluated in 95% CI, and *p* < 0.05 indicated that the differences were statistically significant. Statistical significance was evaluated as two-tailed.

### 2.5. Sample Size Estimate

The primary clinical endpoint was the difference in EGF concentration in breast milk of lactating mothers between the jaundice group and control group. We used SAS 9.4 to estimate the sample size based on a study [19], which showed that the EGF content of breast milk in the BMJ group was 548.36 ± 65.36 pg/mL and that in the control group was 490.26 ± 50.10 pg/mL. The control was matched at a ratio of 1:2 and the sample size was calculated as 48 (α = 0.05, 1 − β = 0.9). Assuming that 30% of the subjects might lose to follow-up up to the date of 42-day postpartum, the sample size of this study was adjusted to 69.

## 3. Results

### 3.1. Participants

A total of 94 maternal and infant pairs were included in this study. Twenty-nine pairs of late-onset BMJ infants and mothers were included in the BMJ group, and 65 pairs of infants without late-onset BMJ and mothers were included in the control group. The comparison of maternal age, BMI, gestational age, mode of delivery, parity, gestational comorbidity, sex, length and weight of newborn, and 1-min Apgar score and 5-min Apgar score between the two groups is shown in Table 1, showing no significant difference.

### 3.2. Comparison of Maternal Diets

There were no differences in maternal diets during puerperium between the BMJ group and control group (*p* > 0.05), except for water intake (*p* < 0.05) (Table 2).

### 3.3. Status of Breastfeeding and Growth

No significant differences were observed in the initial time of lactation, time to start breastfeeding, hospitalization time of puerperae or infant weight loss during hospitalization between the two groups (*p* > 0.05) (Figure 2A).

In the comparison of growth and developmental status between the two groups, there were no differences in neonatal body length, weight, bowel frequency, or breastfeeding frequency at 42 days postpartum, while there was a difference in the delayed second dose of hepatitis B vaccine (*p* < 0.001) (Figure 2B).

### 3.4. Nutrient Concentration in Breast Milk

The concentrations of protein, fat, lactose, minerals, water, density, and energy in breast milk were not significantly different between the BMJ and control groups (*p* > 0.05) (Table 3).

### 3.5. EGF Concentration in Breast Milk

The concentration of EGF in breast milk was significantly lower in the BMJ group (35.47 ± 10.13 ng/mL) than in the control group (42.63 ± 15.36 ng/mL) (*p* = 0.03) (Figure 3).

### 3.6. Association between Breast Milk Concentrations and the TcB Value

Stepwise multiple linear regression was applied to analyze the association between breast milk concentrations, including macronutrients and EGF, and the TcB value. Table 3 shows that the EGF variable was statistically significant both before and after the adjustment. The TcB value decreased by 0.33 (*p* = 0.004) for every 1 ng/mL EGF increase in breast milk in Model 1, after adjusting for maternal age, BMI, gestational age, bowel frequency, breastfeeding frequency, pregnancy complications, newborn birth weight, newborn sex, 1-min Apgar score, 5-min Apgar score, initial time of lactation, time to start breastfeeding, and hospitalization time. EGF and maternal age were found to be negatively associated with late-onset BMJ (*p* < 0.05). The TcB value decreased by 0.27 (*p* = 0.04) for every 1 ng/mL increase in EGF in breast milk, and the TcB value decreased by 0.29 (*p* = 0.03) for every 1 year increase in maternal age in Model 2 (Table 4).

### 3.7. Association between Dietary Food Intake and the TcB Value

The association between dietary food intake and the TcB value was analyzed using multiple linear regression modeling. Before the adjustment (Model 1), cooking oil was found to be statistically significant (*p* = 0.03), and TcB values decreased by 0.27 (*p* = 0.03) for each additional 1 g of oil intake. After adjusting for covariates, soybean, and soybean products (β = −0.34, 95% CI: −2.07, −0.24, *p* = 0.02) intake during lactation was negatively associated with late-onset BMJ, while the cooking oil was no longer statistically significant. (Table 5).

### 3.8. Association between Dietary Food Intake and EGF Concentration in Breast Milk

We used multiple linear regression to investigate the relationship between dietary food intake and EGF concentration in breast milk. Cooking oil was found to be statistically significant in Model 1 (*p* = 0.008). The EGF concentration in breast milk increased by 0.38 ng/mL for each additional 1 g of oil (Table 6).

## 4. Discussion

The aims of this research were to investigate which maternal factors, including postpartum diet and breast milk composition, may influence the occurrence of late-onset BMJ and to identify possible relationships between them. In this study, macronutrients and EGF in mature milk were examined, and the type of food and eating frequency during the postpartum month of lactating mothers were collected. We found that the EGF concentration in breast milk was significantly lower in the BMJ group despite other macronutrients, and the results of regression analysis indicate that lower EGF in breast milk may lead to a higher risk of elevated TcB value. We also observed that the TcB value was negatively correlated with dietary oil intake or soybean and soybean product intake before and after adjustment, respectively. The results of the correlation analysis between EGF and dietary factors in this study suggest that edible oil may affect the development of late-onset BMJ by acting on EGF in breast milk.

The pathogenesis of BMJ is not clear. Most scholars believe that some biological factors in breast milk may lead to an increase in bilirubin hepatoenteric circulation [3]. This case-control study suggests that EGF in breast milk may reduce the risk of late-onset BMJ. EGF is a small peptide growth factor consisting of 53 amino acid residues, with a molecular weight of 6045 Daltons, containing three in-chain disulfide bonds [16]. EGF in the neonatal intestine mainly comes from breast milk, with the highest content in colostrum and a gradual decrease in mature milk [20]. A longitudinal comparison of EGF levels in neonates with late-onset BMJ showed that neonatal serum EGF levels decreased significantly 72 h after the cessation of breastfeeding, while EGF levels in mature breast milk did not change significantly before and after the cessation of breastfeeding, which further suggested that EGF may play a key role in the pathogenesis of late-onset BMJ [17]. Due to immature liver function, the intestinal morphology and microecological environment of infants are unstable, leading to bilirubin metabolism disorder, and bilirubin accumulates in the body and forms jaundice. Animal experiments have shown that EGF in breast milk can promote the growth and maturity of the liver and intestine [21,22,23], thus reducing the content of bilirubin in the blood and inhibiting the occurrence of jaundice.

Our data differ from those of Kumral et al. [10], who suggested that breast milk EGF was positively correlated with serum bilirubin in newborns. Kumral et al. [6] collected milk samples between 3 and 4 weeks postpartum, which was much earlier than ours (6 weeks), so the concentration of EGF in breast milk in Kumral’s study was higher than ours. The collection of breast milk samples at 6 weeks postpartum is more consistent with the characteristics of delayed late-onset BMJ and can compensate for the shortcomings of previous studies. They used TSB (total serum bilirubin), while we used TcB to reflect bilirubin levels, which may partly explain the difference in results.

Previous studies have suggested that the diet, nutrient reserve, and nutrient utilization of lactants may cause changes in bioactive ingredients in breast milk [13,24,25,26,27,28]. Data in our study indicated that the maternal diet may influence the occurrence of late-onset BMJ, and the intake of oil was found to be positively correlated with the concentration of EGF in breast milk. At present, there are a few studies on the effect of lactating mothers’ diets on late-onset BMJ. In this study, the FFQ scale was used for the first time to evaluate the effect of lactating mothers’ diet on the occurrence of late-onset BMJ, and the results showed that the intake of edible oil and soybean and soybean products were negatively correlated with late-onset BMJ before and after adjustment, respectively. Lu M’s study detected the EGF content in breast milk in Beijing, Hangzhou and Lanzhou and investigated the diet during lactation [14]. The results showed that the EGF concentration in breast milk was negatively correlated with the intake of protein, total energy, vegetables, fruits, soy products, and dairy foods. These results suggest that isoflavones and allicin can inhibit the expression of the EGF receptor and reduce the level of EGF in vivo. A study on the effect of walnut oil on wound healing in SD rats [29] showed that walnut oil contains a large amount of unsaturated fatty acids, such as linoleic acid and linolenic acid, which can significantly inhibit NF-κB expression and promote EGF expression. Experiments performed by Bevan et al. [30] on rat liver sections showed that fatty acids, particularly linoleic and linolenic acids, can bind to hepatocyte membrane Z-proteins and thus inhibit bilirubin binding. However, the present study showed that maternal ingestion of soybean and soybean products seems to help reduce TcB concentrations in neonates with late-onset BMJ, but there is no correlation found with the level of EGF in breastmilk. It is reported that maternal intake of soybeans in lactation changed the lipid content of breast milk and programmed offspring for phenotype of the lower metabolic risk [31], which indicated that the effect of soybean on jaundice may be mediated by affecting lipid metabolism of both the mother and the infant, which needs further study. Combining the above findings on the correlation between dietary factors, breast milk EGF and late-onset BMJ, we hypothesized that dietary intake of oil by mothers during the first 6 weeks postpartum may reduce the risk of late-onset BMJ by affecting the level of EGF in breast milk.

It is a common misconception to stop breastfeeding during BMJ. Breastfeeding confers infection protection, superior neurocognition, and maternal cells with capacities for optimal immune direction and pluripotency as well as genetic capabilities. These findings empower breast milk nutrition with broad-based potentials for protection against rare complications that may occur during BMJ [7]. In this case-control study, the second dose of hepatitis B vaccine was significantly reduced in the BMJ group, although we found no significant differences in breast-feeding rates, macronutrient composition, or growth and development at 6 weeks postpartum between the two groups. At present, most community hospitals in Beijing regard BMJ as a contraindication for infant vaccination, leading to delayed vaccination of BMJ infants. Some community hospitals even encourage mothers to suspend breastfeeding and replace it with formula to ensure that the jaundice value drops to the safe range for vaccination, resulting in breastfeeding discontinuation. Recent studies suggest that higher bilirubin levels in infants and children may have beneficial effects on long-term development [32]. Exclusive breastfeeding is clinically and biochemically associated with increased brain development and cognitive abilities that may indicate neurocognitive maturity [5,6]. In addition, the mother–infant psychological connection of exclusive breastfeeding may stimulate tangible emotional circuits that may promote cognitive development [33,34]. If enhanced cognitive development paralleled neurocognitive maturation, breastfeeding indirectly reduced vulnerability to unconjugated bilirubin toxicity. When combined with the holistic knowledge of breastfeeding with BMJ, breastfeeding can provide protection for the immature brain against the adverse effects of jaundice [7]. The Expert Consensus on Vaccination for Children with Special Health Status (Infant Jaundice and Vaccination) recommends that “children with breast milk jaundice who are in good health and have no other complications can be vaccinated following the immunization program”, and regional CDCs (Centers for Disease Control) should moderately relax the criteria for contraindications to vaccination to ensure scheduled vaccination programs and promote breastfeeding [9]. Based on the above evidence, it is recommended that BMJ infants continue to be breastfed and vaccinated on time. At the same time, researchers continue to explore the pathogenesis and prevention of BMJ to minimize its impact on mothers and infants.

There are limitations in breast milk collection, as breast milk is unilateral and the composition of it changes at the beginning and the end of the collection, which may have some impact on the results of the study. Considering the small sample size of this study, further studies focusing on breast milk EGF and BMJ need to be improved by expanding the sample size and incorporating intervention trials.

## 5. Conclusions

EGF in breast milk may inhibit the occurrence of late-onset BMJ. Dietary oil intake of lactating mothers may be a protective factor for the occurrence of late-onset BMJ by increasing EGF levels in breast milk. These results must be confirmed in studies with larger sample sizes and in a wider range of areas. Further studies on the mechanism of the effect of dietary oil intake on EGF content in breast milk need to be further explored. 

## Figures and Tables

**Figure 1 nutrients-14-04587-f001:**
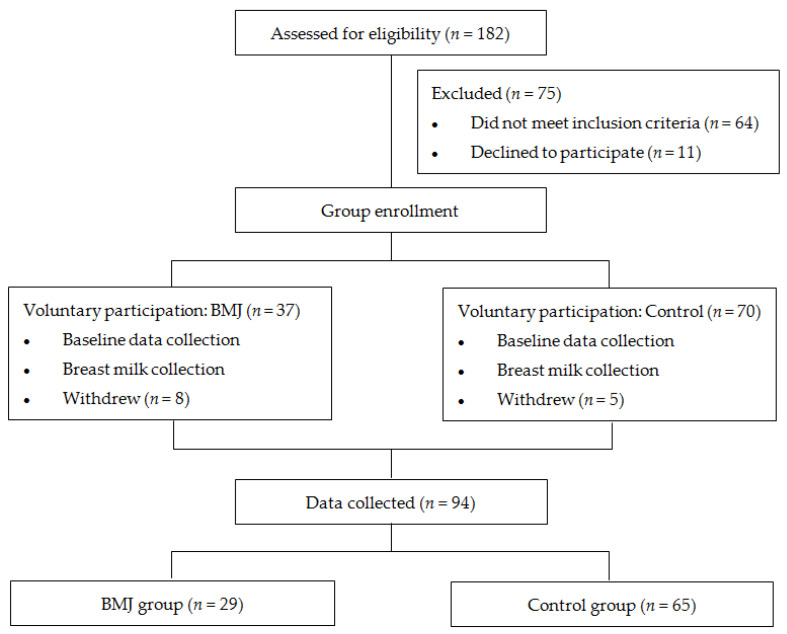
Flowchart.

**Figure 2 nutrients-14-04587-f002:**
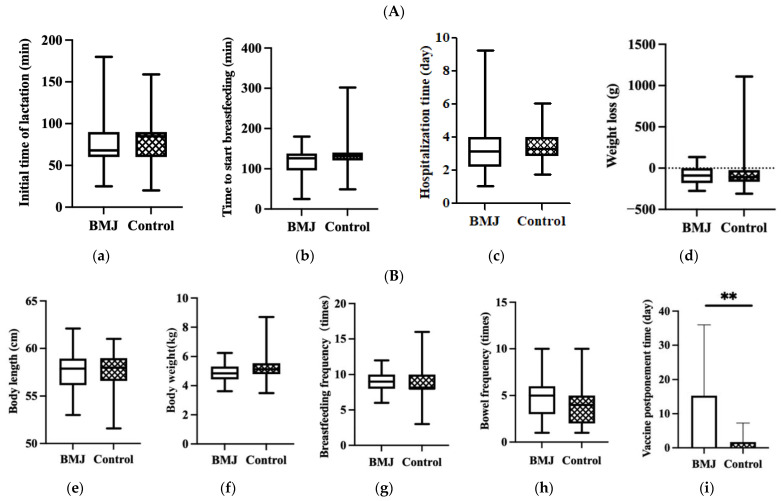
(**A**) Feeding status during hospitalization, (**a**) initial time of lactation, (**b**) time to start breastfeeding, (**c**) hospitalization time, (**d**) weight loss. Error bars represent standard errors. (**B**) Status of growth and feeding of 42-day infants, (**e**) body length, (**f**) body weight, (**g**) breastfeeding frequency, (**h**) bowel frequency, (**i**) vaccine postponement time. Error bars represent standard errors. ** indicates *p* < 0.001.

**Figure 3 nutrients-14-04587-f003:**
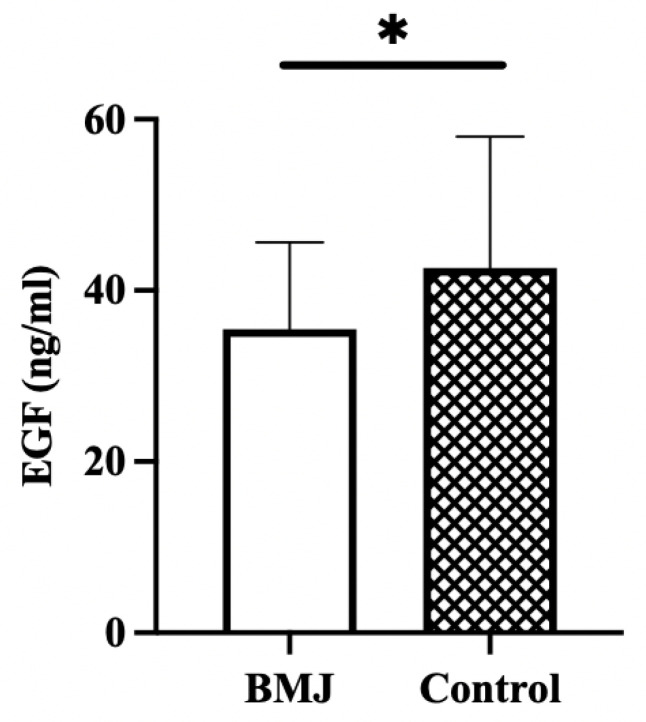
EGF concentration in breast milk. Error bars represent standard errors. * indicates *p* < 0.05.

**Table 1 nutrients-14-04587-t001:** Characteristics of the study population.

Characteristics		BMJ (*n* = 29)	Control (*n* = 65)	χ^2^/t Value	*p* Value
Maternal age(year) ^#^		33.17 ± 4.21	34.40 ± 3.53	−1.45	0.15
BMI ^#^		23.91 ± 3.36	24.30 ± 2.92	−0.57	0.57
Gestational age(week) ^#^		39.14 ± 1.07	39.49 ± 0.87	−1.64	0.11
Delivery ^+^	Vaginal	21 (72.4%)	45 (69.2%)	0.06	0.80
	Cesarean	8 (27.6%)	20 (30.8%)		
Parity ^+^	1	16 (55.2%)	32 (49.2%)	1.37	0.73
	2	8 (27.6%)	22 (33.8%)		
	3	5 (17.2%)	9 (13.9%)		
	4	0 (0.0%)	2 (3.1%)		
Pregnancy complications ^+^	Yes	14 (48.3%)	17 (26.2%)	4.03	0.06
	No	15 (51.7%)	46 (70.8%)		
Infant gender ^+^	Male	17 (58.6%)	32 (49.2%)	0.96	0.33
	Female	12 (41.4%)	33 (50.8%)		
Birth length(cm) ^#^		49.62 ± 2.01	50.41 ± 1.54	−1.87	0.07
Birth weight(g) ^#^		3281.21 ± 457.50	3472.14 ± 460.87	−1.85	0.07
1-min Apgar Score ^#^		9.93 ± 0.26	9.94 ± 0.30	−0.08	0.93
5-min Apgar Score ^#^		9.97 ± 0.19	10 ± 0.00	−1.00	0.33

^#^ mean ± standard deviation. ^+^
*n* (%).

**Table 2 nutrients-14-04587-t002:** Comparison of maternal dietary nutrients between BMJ group and control group.

	Mean ± SD		t Value	*p* Value
	BMJ Group	Control Group		
Energy (kcal/d)	1863.34 ± 904.98	2261.43 ± 1443.45	−1.045	0.303
Protein (g/d)	80.79 ± 49.98	91.60 ± 52.13	−0.670	0.507
Fats (g/d)	84.50 ± 22.85	113.98 ± 89.81	−1.423	0.163
Carbohydrates (g/d)	202.85 ± 166.21	226.78 ± 136.69	−0.497	0.622
Dietary fiber (g/d)	11.47 ± 5.85	13.78 ± 5.68	−1.268	0.213
Calcium (mg/d)	624.20 ± 279.41	944.78 ± 678.33	−1.954	0.058
Iron (mg/d)	19.90 ± 10.75	24.58 ± 13.84	−1.196	0.239
Zinc (mg/d)	12.12 ± 8.67	14.06 ± 9.07	−0.693	0.493
Selenium (μg/d)	51.10 ± 27.11	64.81 ± 36.80	−1.342	0.188
Copper (mg/d)	1.95 ± 1.12	2.07 ± 0.81	−0.387	0.701
Manganese (mg/d)	3.87 ± 6.14	3.53 ± 2.36	0.236	0.815
Magnesium (mg/d)	356.72 ± 258.71	364.53 ± 184.73	−0.110	0.913
Sodium (mg/d)	2825.82 ± 656.49	3238.30 ± 2128.54	−0.828	0.416
Potassium (mg/d)	2440.76 ± 1116.17	2737.27 ± 111965	−0.839	0.407
Phosphorus (mg/d)	1121.04 ± 596.07	1388.09 ± 815.63	−1.182	0.244
Iodine (μg/d)	87.02 ± 96.53	88.98 ± 73.47	−0.072	0.943
Vitamin A (μgRAE/d)	982.92 ± 487.66	1275.13 ± 621.87	−1.654	0.106
Vitamin E (mg/d)	28.53 ± 7.84	29.68 ± 8.84	−0.436	0.665
Vitamin B1 (mg/d)	1.00 ± 0.52	1.24 ± 0.75	−1.206	0.235
Vitamin B2 (mg/d)	1.30 ± 0.50	1.83 ± 1.11	−1.975	0.056
Vitamin C (mg/d)	94.00 ± 48.70	120.89 ± 55.26	−1.633	0.111
Niacin (mgNE/d)	15.89 ± 11.09	18.66 ± 16.33	−0.627	0.535
Folate (μg/d)	378.35 ± 165.63	465.55 ± 230.80	−1.373	0.178
Water (g/d)	861.59 ± 302.19	1233.46 ± 662.91	−2.283	0.028

**Table 3 nutrients-14-04587-t003:** Comparison of nutrient concentrations in breast milk between BMJ group and control group.

	Mean ± SD		t Value	*p* Value
	BMJ Group	Control Group		
Protein (g/100 mL)	0.887 ± 0.23	0.979 ± 0.16	−1.473	0.149
Fat (g/100 mL)	4.167 ± 1.30	4.10 ± 0.74	0.194	0.848
Lactose (g/100 mL)	7.239 ± 1.86	7.98 ± 1.35	−1.454	0.154
Minerals (g/100 mL)	0.217 ± 0.06	0.240 ± 0.04	−1.537	0.133
Water (g/100 mL)	87.49 ± 2.07	86.70 ± 1.82	1.293	0.204
Density (g/100 mL)	1.025 ± 0.01	1.028 ± 0.01	−1.432	0.160
Energy (kcal/100 mL)	70.00 ± 11.67	72.77 ± 9.67	−0.818	0.419

**Table 4 nutrients-14-04587-t004:** Association between the composition of breast milk and the TcB value.

	Model 1 ^a^				Model 2 ^b^		
	β	95% CI	*p*		β	95% CI	*p*
EGF	−0.33	(−0.16, 0.03)	0.004	EGF	−0.27	(−0.16, −0.00)	0.04
—				Maternal age	−0.29	(−0.55, −0.03)	0.03

^a^. Independent variables included minerals, protein, density, fat, lactose, energy, water and EGF. Only selected variables are shown in the table. ^b^. In addition to the variables included in model 1, maternal age, BMI, gestational age, bowel frequency, breastfeeding frequency, pregnancy complications, newborn birth weight, newborn gender, 1-min Apgar score, 5-min Apgar score, initial time of lactation, time to start breastfeeding, and hospitalization time were added in Model 2. Only the variables finally included in the model after screening are shown in the table.

**Table 5 nutrients-14-04587-t005:** Association between dietary food intake and the TcB value.

Food	Model 1 ^a^			Food	Model 2 ^b^		
	β	95% CI	*p*		β	95% CI	*p*
Cooking oil	−0.27	(−6.43, −0.38)	0.03	Soybean and soybean products	−0.34	(−2.07, −0.24)	0.02

^a^. Independent variables included milk, cereal, tubers, soybean and soybean products, vegetable, fruit, meat, egg, and cooking oil. Only selected variables are shown in the table. ^b^. In addition to the variables included in model 1, maternal age, BMI, gestational age, bowel frequency, breastfeeding frequency, pregnancy complications, newborn birth weight, newborn gender, 1-min Apgar score, 5-min Apgar score, initial time of lactation, time to start breastfeeding, and hospitalization time were added in Model 2. Only the variables finally included in the model after screening are shown in the table.

**Table 6 nutrients-14-04587-t006:** Association between EGF and dietary food intake.

EGF	Model 1 ^a^		
	β	95% CI	*p*
Cooking oil	0.38	(5.75, 35.36)	0.008

^a^. Independent variables included milk, cereal, potato, soybean and soybean products, vegetable, fruit, meat, egg, and oil. Only selected variables are shown in the table.

## Data Availability

Not applicable.

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
