# Peer review of "Effect of Epidermal Growth Factor in Human Milk and Maternal Diet on Late-Onset Breast Milk Jaundice: A Case-Control Study in Beijing"

_nutrients, 2022, doi:10.3390/nu14214587_

Round 1
Reviewer 1 Report (Previous Reviewer 2)
no more comments
Author Response
Thank you for your review.
Reviewer 2 Report (New Reviewer)
The manuscript "Effect of epidermal growth factor in human milk and maternal diet on late-onset breast milk jaundice: a case-control study in Beijing" is one of the numerous articles in the field of late-onset BMJ.
This study aims to find a connection between a mother's milk composition and late-onset BMJ. The authors showed the association between maternal diet and EGF on the late onset of BMJ.
The severity of jaundice was measured by the TcB method, and the EGF by the ELISA method.
The composition of milk depends on the mother's diet. The level of EGF mainly depends on the type of diet and the amount of energy consumed. In this study, the authors found a correlation between the composition of breast milk and the late onset of BMJ. Is this proof of EGF's impact on BMJ?
Recommendations:
· The level of EGF mainly depends on the type of diet and the amount of energy consumed. I recommend shortening the title: Effect of maternal diet on late-onset breast milk jaundice: a case-control study in Beijing.
· The abstract is unnecessarily long, so it should be shortened according to the journal's recommendation.
· In addition to SPSS, the authors also used another program for graphic display. It should be added name of the user program to the statistical analysis. Line 106 Flowchart: in the BMJ group of 29, 8 withdrew, and in the control group of 65, 5 withdrew. In the results, the number in the BMJ group is 29, and in control 65. Please explain.
· The results in figure 3 are challenging to follow. Show the results in a table.
Author Response
Please see the attachment.

This manuscript is a resubmission of an earlier submission. The following is a list of the peer review reports and author responses from that submission.
Round 1
Reviewer 1 Report
Thanks to the authors for this work. A very interesting and well developed article that studies the relationship between epidermal growth factor (EGF) levels in breastmilk and the association with neonatal jaundice. In addition, it postulates foods for maternal consumption that could improve EGF levels in their brestmilk. To improve the article, I have some suggestions and notes that I would like to discuss with the authors.
1. The introduction is very well described. It is concise and concrete and the proposal of your study is well understood. However, I do not think that the objective "to explore possible mechanisms" can be covered in a case-control study. After reading the article in depth, they could refer to nutritional strategies that improve EGF levels in breast milk.
2. In the material and methods again it is very understandable, the section on sample size calculation is extremely explanatory, but wasn't some kind of % of losses considered in the sample size calculation?
2.1. why were there 8 and 5 final withdrawals per group?
2.2. why was 42-day postpartum chosen to collect the data?
2.3. Although I agree with the way in which the milk sample is collected, it could infer limitations, since it is unilateral, it is not a 24h pool, even knowing that it changes at the beginning and at the end of the collection. This should be described in the discussion.
3. Results: I think they are very well understood, and the development of the results is easy to follow, but,
3.1. the dispersion of the control group in all variables is huge, why is this?
3.2. weight and length could be reported in Z-scores for gestational age and sex? these scores could be more comparable with the rest of the populations.
3.3. After reviewing the models, the adjustment variables were always introduced and the variables such as maternal age, BMI, etc, were one-by-one introduced?, if these were significant, were they kept together with the level of milk EGF. If this is true, perhaps the authors should explain the construction of the models in the statistical section.
On the other hand, if the adjustment variables were not significant in the univariate analysis, it could choose the most relevant ones to adjust the models, but not all of them (considering their sample size).
4. In the discussion,
4.1. Have correlations been observed (lines 236-239)?
4.2. Although it appears that EGF levels are stable, I think it should be reported that these data have been found in mature milk, and should be corroborated at earlier stages (lines 250-252), which leads me to the next question,
4.3. The explanation of their data in relation to Kumral et al study, makes me think that if EGF levels in breastmilk could become a biomarker? if so, should they be measured from 4-5 months postpartum?
4.4. The discussion about unsaturated fatty acids is very interesting. What type of oil is the most consumed in their population, is it more olive oil (high oleic, palmitic?) or more corn oil (LA content?).
5. Strongly agree with the conclusions reported.
Minor comments:
- Review the author guidelines, there are some type errors (line 88, what is a "psycho-psychological" disorders?, line 130 non-normal rather than "nonnormally", line 131 similar). Also, check the way references would be cited.
- Keywords: write BMJ and EGF meaning. "Diet" is a very broad term, perhaps it could be refined a little more to make the search terms more precise.
- line 79-80, move it to the introduction.
- Define BMJ diagnostic in the material and methods.
- Try to describe the "control" group as control and not as "normal" (line 148).
- Table 1, "t value" refers to a non-parametric test? (as described in the statistics section).
Reviewer 2 Report
The findings of this study are different from traditional perceptions and need to be more conservative in interpretation and explanation.
There are some suggestions as follows.
1. In the introduction section, the possible causes of late-onset breast milk jaundice in the newborn should be described.
2. Please describe how many days after birth the late-onset BMJ cases occurred? Did all cases continue to breastfeed after the occurrence?
3. Line 103 must describe which food items are included in the food frequency questionnaire (FFQ). I am confused about the food items presented below table 3. Why only potato is listed, are others such as pumpkin and sweet potato evaluated? Does the oil refer to cooking oil? Whether plant and animal sources are included or not? The fatty acid composition differs greatly.
4. The representation of the values in Table 1 are incorrect.
In the Delivery section, among the 29 BMJ cases, 21 were vaginal, accounting for 72.4% of the BMJ cases, and 8 were cvesarean, accounting for 27.6% of the BMJ cases. Among the 65 control cases, 44 are vaginal, accounting for 67.7% of the cases in the control group, and 19 are cvesarean, accounting for 29.3% of the cases in the control group. Please modify the entire table according to this concept.
5. Please add error bar in Figure 3
6. The title of table 2 "Association between breast milk concentrations and late-onset BMJ." is suggested to be revised to "Association between the composition of breast milk and the TcB value."
7. The title of table 3 “Association between dietary food intake and late-onset BMJ.” is suggested to be revised to " Association between dietary food intake and the TcB value."
8. In the table3 the item of oil was not clear, please specify that it was cooking oil or total dietary fat. Please indicate whether oil still has statistical difference after adding other variables (model 2)
9. Why look at the correlations only by food group and not by overall nutrient intake? A study mentions the correlation between fatty acid composition in breast milk and breast milk jaundice for your reference. https://pubmed.ncbi.nlm.nih.gov/33327994/
10. The present results indicated that dietary oil was associated with EGF. Some studies also pointed out that dietary fat can affect the content and composition of breastmilk fat. Is breastmilk fat also was associated with EGF?
11. This result indicates that maternal ingestion of soybean and soybean products seems to help reduce TcB concentrations in neonates, but there is no correlation with EGF, please describe the possible mechanism.
12. Late-onset breast milk jaundice may be related to the inability of the newborn's liver to metabolize some components of breast milk and usually gets better after stopping breastfeeding. Considering the benefits of breastfeeding, increasing EGF in breast milk may help improve jaundice index without stopping breastfeeding.
